# Flow–Solid Coupling Analysis of Ice–Concrete Collision Nonlinear Problems in the Yellow River Basin

Li Gong [1,2], Zhouquan Dong [1,*], Chunling Jin [1], Zhiyuan Jia [1] and Tengteng Yang [1]

1 Department of Civil Engineering, Lanzhou Jiaotong University, Lanzhou 730070, China
2 Institute of Water Diversion Engineering and Security of Water Transferring, Lanzhou Jiaotong University, Lanzhou 730070, China
* Correspondence: dzq2416796947@163.com

**Abstract:** Yellow River ice is the most prominent and significant natural disaster in winter and spring in China. During the drift ice period, water transmission tunnels located in this area tend to be hit by water–drift ice coupling. Thus, it is an important issue to reduce water transmission tunnel damage by drift ice, ensure the safety of operation and maintenance, and prevent engineering failure. In this paper, a numerical simulation of the collision process between ice and the tunnel is carried out by using the fluid structure coupling method and ANSYS/LS-DYNA finite element software. In addition, a model test with a geometric scale of 1:10 is carried out to verify the numerical simulation results, and the mechanical properties and damage mechanism of drift ice impacting the tunnel concrete lining in water medium are studied. The results show the following: the experimental values of maximum equivalent stress and *X*-directional displacement of the flow ice on the water transfer tunnel have the same trend as the simulated values, both of which show an increasing trend with an increase in flow ice velocity. It is shown that the ice material model parameters, ALE algorithm, and grid size used in this paper are able to simulate the impact of drift ice on the water transfer tunnel more accurately. With an increase in drift ice collision angle and drift ice size, the fitted curves of equivalent stress and peak displacement in *X*-direction all show relationships of exponential function. The peak value of displacement in the *X*-direction and maximum equivalent stress decrease with an increase in the curvature of the tunnel structure. It is also shown that the influence of change in drift ice size on the tunnel lining is greater than that of a change in tunnel section form. It is found that a high-pressure field will be formed due to extrusion of flowing ice, which should be fully considered in the numerical simulation. The research method and results can provide technical reference and theoretical support for prevention and control of ice jam disasters in the Yellow River Basin.

**Keywords:** water medium; numerical simulation; model test; fluid–solid coupling; drift ice; water diversion tunnel; impact effects





## 1. Introduction

The Yellow River is the second largest river in China, located north of the Tropic of Cancer and in the shape of "several" [1]. Its special geographical location, climatic and hydrological conditions, river topography, and the flow direction of part of the river from low latitudes to high latitudes lead to frequent ice disasters, posing a great challenge to safe operation of local hydraulic structures [2]. The water transfer tunnel located in this region is often hit by drift ice during the ice flow period, and, over the long term, collisions will cause damage and failure of its concrete lining, which will affect the service life of the water transfer tunnel to a certain extent [3]. Therefore, there is an urgent need for research on the impact of small- and medium-sized drift ice on water transfer tunnels in the Yellow River Basin [4].

In order to reasonably allocate water resources and alleviate water conflicts between different regions, China has built many water transfer projects, and water transfer tunnels

have become one of the most important forms of water transfer [5]. Water transmission tunnels generally work under pressure and free surface conditions, and this mode of operation involves them in fluid–structure interaction, soil/rock–structure interaction, etc. Xie et al. [6] studied the variation law of water pressure outside tunnel lining. Simanjuntak et al. [7], for the internal water pressure to which a pressure tunnel is subjected, estimated distribution of seepage pressures and water losses around concrete-lined pressure tunnels pre-stressed by grouting. In addition, concrete lining structures are widely used due to the special operating conditions of water transmission tunnels and considering the economics and construction technology. However, internal transient pressure during operation can lead to cracks in concrete lining, which can further lead to lining leakage and engineering failure. To address this engineering problem, Mehrdad [8] used numerical simulation software to model and analyze a pressure tunnel and studied the ultimate bearing capacity of tunnel concrete lining. Pachoud et al. [9] studied SIF for axial semi-elliptical surface cracks and embedded elliptical cracks at longitudinal butt-welded joints of steel liners by means of the finite element method. Evidently, most of the above studies have focused on the effects of internal and external pressures, etc., on tunnels, while the interaction between small- and medium-sized drift ice and water transfer tunnels has been less studied.

Scholars at home and abroad have conducted significant research on ice–structure and ice–ship collisions. The research tools are mainly in situ observations, physical tests, and numerical simulations. Although the results of field observations and physical tests are highly reliable and can provide standards for numerical simulations, experimental research is costly and time-consuming and only a small amount of data can be obtained. In recent years, numerical simulation has been widely used with development of computer technology and has significant advantages regarding ice–ship and ice–structure collision processes, which have been studied by several scholars. A three-dimensional ice intrinsic model with nonlinear viscoelastic and plastic components in series was proposed by Xu et al. [10] and used for dynamic simulations of ice–structure interactions. Li et al. [11] conducted a qualitative study of ice damage through model observations and the extended finite element method (XFEM). Interaction pattern between drift ice and open channels has been studied by Gong [12] and others. Kim [13] established a simulation model of a fixed structure and ice using the discrete element method and numerically simulated the wear process of structure and ice using the finite element method. Hayo et al. [14] investigated ice-excited vibrations with numerical simulation software after considering ice floe size and wind and water current factors. Wang et al. [15] established a numerical simulation method for ice–water–structures based on the structured arbitrary Lagrangian Eulerian (S-ALE) method. Yu et al. [16] proposed a numerical solver based on a subroutine in LS-DYNA software to simulate collision of ice floes with offshore structures and discussed ice break-up and damage distribution of structures under different degrees of freedom. Wang et al. [17] innovatively applied the state-based near-field dynamics (PD) method to the sea–ice impact problem and analyzed the various sensitivity factors in the sea–ice impact process. However, study of numerical simulation of solid–fluid–structure is still in its infancy as it involves two-phase flow motion and fluid–solid interactions. Istrati [18] used CFD methods for numerical investigation of tsunami-borne debris damping loads on a coastal bridge. Pasculli [19] used the SPH method for modeling of fast muddy debris flows. Hasanpour [20] innovatively used SPH–FEM modeling to investigate the impact of tsunami-propagated large debris flows on coastal structures and further investigated debris–fluid–structure interactions [21]. Istrati [22] further developed a numerical simulation of the effects of large debris on bridge piers using a multi-physics field complex modeling approach. In summary, scholars at home and abroad have conducted a great deal of research on ice–ship, ice–structure, and solid–fluid–structure collisions, but research on the collision mechanism of small-and medium-sized drift ice on concrete structures with consideration of water medium is still in its infancy.

Therefore, in order to investigate the mechanical properties and damage mechanism associated with small to medium drift ice on concrete lining of water transfer tunnels

during the drift ice period of the Yellow River Basin and to understand the influence of water medium on ice–tunnel collisions, this study adopts the fluid–solid coupling method combined with model test to analyze the factors affecting tunnel life and discover the impact of influence law of flow ice on water transmission tunnels under different working conditions, which is intended to provide theoretical support and technical reference for prevention and control of ice disasters in the Yellow River Basin.

## 2. Numerical Simulation

### 2.1. Display Time Integration Principle

The impact of drift ice on a tunnel is a transient dynamic response process that occurs over a very short period of time. The drift ice near the collision zone produces deformation, overturning, breakage, and other phenomena, the tunnel collision zone appears in a high-stress area, and displacement changes. Therefore, in order to explore the nonlinear dynamic response law in the ice–tunnel collision process, based on the display time integration method, according to Newton's law, after considering the damping hourglass, the dynamic equation of the ice–tunnel collision process is as shown in (1):

$$M\ddot{x}(t) = P(t) - F(t) + H(t) - C\dot{x}(t) \tag{1}$$

where: $M$ is the mass matrix; $C$ is the viscous damping matrix; $\ddot{x}(t)$ and $\dot{x}(t)$ are the nodal acceleration and velocity vectors at moment $t$, respectively; $P(t)$, $F(t)$, and $H(t)$ are the load vector, internal force vector, and hourglass resistance vector, respectively. The load vector and internal force vector in Equation (1) are calculated by the following two equations:

$$P(t) = \sum_e \left( \int_{Ve} N^T f dV + \int_{\partial b_{2e}} N^T \overline{T} dS \right. \tag{2}$$

$$F(t) = \sum_e \int_{Ve} B^T \sigma dV \tag{3}$$

where: $f$ is the body force vector; $\overline{T}$ is the surface force vector; and $\partial b_{2e}$ is the stress boundary condition.

For acceleration, velocity and displacement can be solved recursively by the central difference method, which has the following recursive format.

$$\ddot{x}(t) = M^{-1}\left[P(t) - F(t) + H(t) - C\ddot{x}(t_{n-1/2})\right] \tag{4}$$

$$\dot{x}(t_{n+1/2}) = \dot{x}(t_{n-1/2}) + \ddot{x}(t_n)(\Delta t_{n-1} + \Delta t_n)/2 \tag{5}$$

$$x(t_{n+1}) = x(t_n) + \dot{x}(t_{n+1/2})\Delta t_n \tag{6}$$

### 2.2. Fluid–Solid Coupling Method

Based on the ice tunnel collision scene in the water medium described in this paper, considering the material nonlinearity, contact nonlinearity, and the presence of prominent transient mutation characteristics, and a variety of physical processes involved in the collision process, the "virtual collision" between the ice and tunnel was conducted by the ALE (arbitrary Lagrange–Euler) algorithm. In the ALE algorithm, the Euler algorithm and Lagrangian algorithm are used to describe fluids (water, air) and solids (ice floe, tunnel), respectively. Finally, fluids and solids are coupled together by the keyword * CONSTRAINED_LAGRANGE_IN_SOLID to realize coupling of water–air–and ice floe–tunnel.

The algorithm control equations include the following equations.
Conservation of mass equation:

$$\frac{\partial \rho}{\partial t} + \rho \nabla \cdot v + (v - w) \cdot (\nabla \rho) = 0 \tag{7}$$

Conservation of momentum equation:

$$\rho\frac{\partial v}{\partial t} + \rho[(v - w) \cdot \nabla)]v = \nabla\sigma + \rho g \tag{8}$$

Conservation of energy equation:

$$\rho\frac{\partial e}{\partial t} + \rho[(v - w) \cdot \nabla)]e = (\sigma : \nabla)v + \rho g \cdot v \tag{9}$$

where $\rho$ is the material density, $v$ is the velocity of the material mass; $w$ is the velocity of the grid; $\sigma$ is the stress tensor; $g$ is the force acceleration; $e$ is the energy; and $t$ is the time.

## 3. Modelling of Tunnel Impact by Drift Ice

### 3.1. Engineering Examples

In this paper, a section of Pandaoling Tunnel 37# of the "Ying Da Into Qin" project in Northwest China was selected as a prototype for modelling. The length of the tunnel selected for this study is 15.723 km, the flow rate is 36 m$^3$/s, and the tunnel has a clear height of 4.40 m, a clear width of 4.20 m, and a vault radius of 2.10 m.

### 3.2. Selection of Model Material Parameters

(1) Drift ice material model: At present, the research on ice materials is mainly focused on sea ice, while the research on river ice materials is relatively scarce. The drift ice in the river shows different mechanical properties under conditions of different temperature and strain rate. In this paper, the ice material is combined with the uniaxial compression test of river ice carried out in [23]. The plastic material related to strain rate (* MAT_STRAIN_RATE_DEPENDENT_PLASTICITY) is selected to simulate the characteristics of drift ice. The parameters of ice material model are shown in Table 1.

**Table 1.** Ice material parameters.

| Parameters | Values |
|:---:|:---:|
| Density (kg·m$^{-3}$) | 910 |
| Modulus of elasticity/GPa | Related to strain rate |
| Poisson's ratio | 0.3 |

(2) Tunnel material model: In the process of ice–tunnel collision, the tunnel lining will be damaged and deformed, so selection of tunnel lining materials should undergo plastic deformation. The concrete material model is CSCM-CONCRETE model developed by FHA Company [24]. The parameters used in the concrete material model are shown in Table 2.

**Table 2.** Parameters of the concrete material model.

| Parameters | Values |
|:---:|:---:|
| Mass density/(kg·m$^{-3}$) | 2500 |
| Maximum strain increment | 0 |
| Rate effect switch | 1 |
| Pre-existing damage | 0 |
| Erosion coefficient | 1.1 |
| Coefficient recovery parameter | 10 |
| Blocking options | 0 |
| Compression strength/MPa | 29 |
| Aggregate size/m | 0.02 |

(3)    Water and air media models: The constitutive model and equation of state are often used in LS-DYNA software to describe fluid materials (water, air). Therefore, the blank material Null was chosen to simulate water and air [25] with the parameters shown in Table 3. The equation of state is defined by the polynomial equation and the Gruneisen equation [26] for water and air media, respectively, and the parameters of both equations of state are shown in Table 3.

**Table 3.** Water and air material model parameters.

| Parameters | Water | Air |
|---|---|---|
| Density/$(kg/m^3)$ | 1.1845 | 998.21 |
| Cutting stress/(Pa) | −10 | $-1 \times 10^5$ |
| Viscosity coefficient | $1.7456 \times 10^{-5}$ | $8.684 \times 10^{-4}$ |
| Constant $C$ | | 1647 |
| Constant $S_1$ | | 1.921 |
| Constant $S_2$ | | −0.096 |
| Constant $C_4$ | 0.4 | |
| Constant $C_5$ | 0.4 | |
| Constant | | 0.35 |
| Initial internal energy$(E_0/J)$ | $2.53 \times 10^5$ | $2.895 \times 10^5$ |
| Initial internal energy/$V_0$ | 1.0 | 1.0 |

*3.3. Model Building*

The collision between drift ice and a water transfer tunnel is a complex dynamic response process. Considering the efficiency of the solution and the accuracy of the calculation and the fact that the tunnel is a symmetrical structure, this paper establishes half of the tunnel for the solution and analysis based on the actual project. As the tunnel is generally in a water body environment, when it is hit by drift ice, the presence of the surrounding water medium will have a damping effect on the structure, which not only affects the dynamic response characteristics of the tunnel structure but also, when the drift ice hits the tunnel at a certain speed, the fluid dynamic load will impact the movement of drift ice through the coupling algorithm, while the movement of the drift ice will cause energy changes in the fluid. Therefore, in this paper, the effects of the additional mass model without considering the water medium and fluid–solid coupling model considering the water medium are explored separately. The principle of the additional mass method is to ignore the water medium during modelling and attach the dynamic effect of the water medium to the drift ice in the form of additional mass to improve calculation accuracy while reducing modelling time. At present, there are many studies on ship–ice and ship–ship collision by additional mass method. Motora [27–29] found that the additional mass $m$ of the impacting motion hull is small compared with the hull mass $m_o$, only accounting for 2–7% of the hull mass, as, the longer the impact time is, the larger the additional mass is, while the ice–tunnel impact time is very short. Therefore, in this paper, the additional mass coefficient was set at 0.02 to carry out the additional mass of drift ice $\Delta m$ calculation in the simulation by adjusting the drift ice density parameters to change the unit volume of drift ice mass [30], as well as through Equation (10) to complete additional mass of the conversion calculation. The fluid–solid coupling model, i.e., considering the dynamic load effect of the water medium on the tunnel lining structure in the actual project, was used to establish the water and air domains, respectively. Second, in order to improve the solution efficiency, only the influence of the water medium on the collision in the collision region was considered. Therefore, a $4 \times 1.6 \times 0.75$ m$^3$ air domain model and a $4 \times 1.6 \times 2.25$ m$^3$ water domain model were selected. The air domain is not selected as the full domain, but a layer of 0.75 m thickness is selected for calculation and analysis because, by observing the natural drift ice, most of the floating drift ice is immersed in the water medium, only a small part is in the air medium, and this paper studies the impact of small- and medium-sized drift ice on the collision of the water transfer tunnel, so the impact of the air domain on

the drift ice is small, and, based on the flow–solid coupling method used in this paper for calculation and analysis, if the Eulerian grid is too much, the computation time of fluid–solid coupling will be greatly increased. Therefore, this paper selects part of the air domain for calculation and analysis of the whole model under the condition of ensuring the accuracy of the collision results.

$$\rho = (1 + m_x)\rho_o \tag{10}$$

where: $\rho$ is the mathematically converted drift ice density in kg/m$^3$; $m_x$ is the additional mass factor; and $\rho_o$ is the initial drift ice density in kg/m$^3$.

The contact algorithm used for the collision between drift ice and the water-conveyance tunnel is based on the symmetric penalty function method, mainly because it has the advantage of being symmetrical and accurate and can also effectively reduce the hourglass effect. The contact type is automatic surface to surface contact (ASSC) [31], where the tunnel lining surface is the dominant surface and the drift ice is the slave surface. In finite element numerical simulations, the tunnel, drift ice, water medium, and air are all solid 164 cells with single point integration to avoid volume locking of the cells, while virtual hourglass stiffness is introduced to prevent possible zero energy patterns during the ice–tunnel collision. A reasonable choice of grid size can achieve a dynamic balance between computational accuracy and efficiency because a smaller grid size can produce convergent computational results to improve the reliability of the numerical simulation results. Second, in order to improve the computational efficiency, the grid size cannot be too small; therefore, this study used VSWEEP to divide the grid. The tunnel, water, and air domain grid sizes are all 0.1 m, and the drift ice grid size is determined to be 0.05 m through a grid size sensitivity study. In addition, the boundary conditions and initial conditions of the Eulerian cells needed to be defined in the fluid–solid coupling model, and reflection-free boundary conditions were set on both sides of the water and air domain along the z-direction to simulate an infinite basin to ignore the influences of reflected waves. In order to keep the motion of the drift ice in the water–air coupled medium unaffected, the degrees of freedom of the water domain along the *x*-positive direction were constrained and the rest of the domain was defined as the free access boundary of the fluid. For the tunnel boundary conditions, we here apply full constraints on the tunnel lining floor and side walls.

In order to accurately simulate the impact of flowing ice on the water conveyance tunnel in the water medium, the energy consumption of the water medium was reduced, and then more real collision results were obtained. In the calculation of the ice–tunnel collision angle, the forward collision form was adopted to reduce the blocking effect of the water medium. Considering that the impact of ice on the tunnel lining is mainly in the *X*-direction in practical engineering, the initial velocity of ice was in the *X*-direction in this simulation to provide power for the ice. In order to reduce the calculation time, the model was simplified, thus ignoring wind and air temperature. Second, in order to avoid the penetration phenomenon at the beginning of the collision and ensure small energy loss during movement of the drift ice, the distance between the drift ice and the tunnel in the *X*-direction was set to 0.005 m in the simulation. The additional mass diagram and flow−solid coupling model diagram of the ice-tunnel collision are shown in Figures 1 and 2.

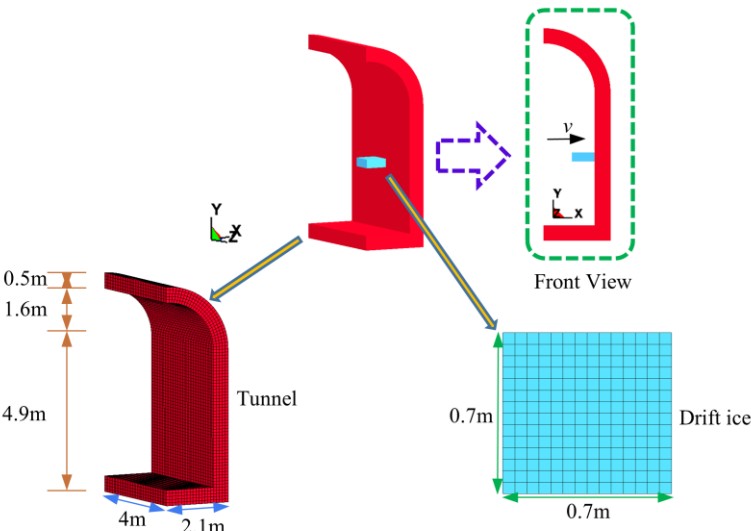

**Figure 1.** Additional mass model.

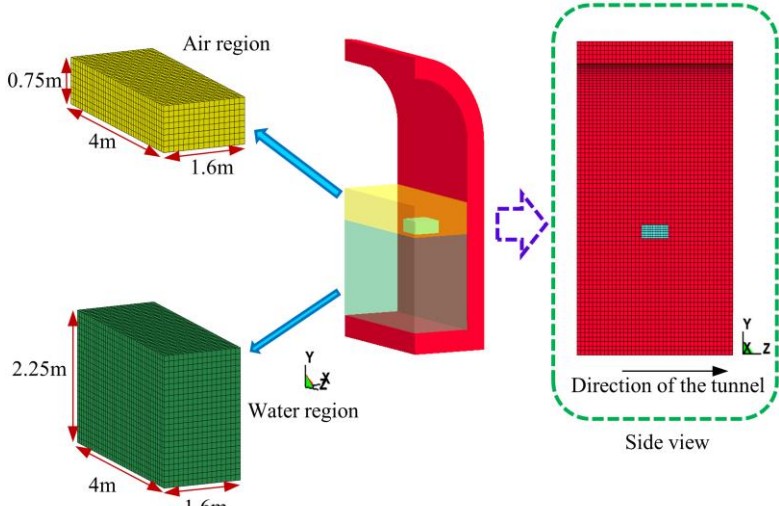

**Figure 2.** Fluid–solid coupling model.

### 3.4. Mesh Sensitivity Study

In order to study the influence brought by the flow ice grid scale on the numerical results, three sets of grids A1, A2 and A3 were selected with corresponding dimensions of $0.05 \times 0.05 \times 0.05$, $0.08 \times 0.08 \times 0.08$ and $0.10 \times 0.10 \times 0.10$ m$^3$, and the peak equivalent stress and peak displacement in the *X*-direction derived from the numerical calculations at different grid scales are given in Table 4.

**Table 4.** Comparison of peak equivalent stress and peak displacement in *X*-direction at different mesh scales.

| Mesh Serial Number | A1 | A2 | A3 | Experimental Value |
|---|---|---|---|---|
| mesh size | $0.05 \times 0.05 \times 0.05$ m$^3$ | $0.08 \times 0.08 \times 0.08$ m$^3$ | $0.10 \times 0.10 \times 0.10$ m$^3$ | |
| Peak equivalent stress ($\times 10^6$ Pa) | 2.475 (2.1%) | 2.916 (−15%) | 3.211 (−27%) | 2.528 |
| *X*-direction displacement peak ($\times 10^{-5}$ m) | 5.216 (0%) | 5.913 (−13%) | 6.295 (−21%) | 5.216 |

From Table 4, we can see that the errors of the experimental values of the peak equivalent stress obtained from the simulations of A1, A2 and A3 meshes are 2.1%, −15% and −27%, respectively; the errors of the peak displacement in the *X*-direction obtained from the simulations of A1, A2 and A3 meshes and the experimental values are 0%, −13% and −21%, respectively. Through the above mesh size sensitivity analysis, the A1 ($0.05 \times 0.05 \times 0.05$ m$^3$) mesh is used in this paper for the following numerical calculations.

### 3.5. Test Verification

In this paper, eight ice–tunnel collision scenarios were established to verify the accuracy of the numerical simulation results under eight flow velocity conditions, such as 0.6, 1.2, 1.8, 2.4, 3.0, 3.6, 4.2, 4.8 m/s, based on the flow velocity information of the diversion into Qin. Inertial forces play a major role when drift ice strikes the tunnel lining at a certain speed and mass (change in mass due to change in dimensions), while elastic forces play a major role when the tunnel lining is deformed elastically by impact of drift ice. Therefore, the Froude and Corsi criteria [32] were used, respectively, for these two types of problems. Geometric ratio C1 between the indoor model tests and the actual model was determined to be 1:10, with a material density ratio of 1.0 and an acceleration ratio of 1.0, and the tests were carried out under normal gravity field conditions using a positive touch form and C30 tunnel lining for results verification.

The ice–concrete collision test system was used to complete determination of strain in the collision zone of the tunnel lining and verify the simulation results. As shown in Figure 3, the ice–concrete collision test set-up mainly consisted of the following equipment.

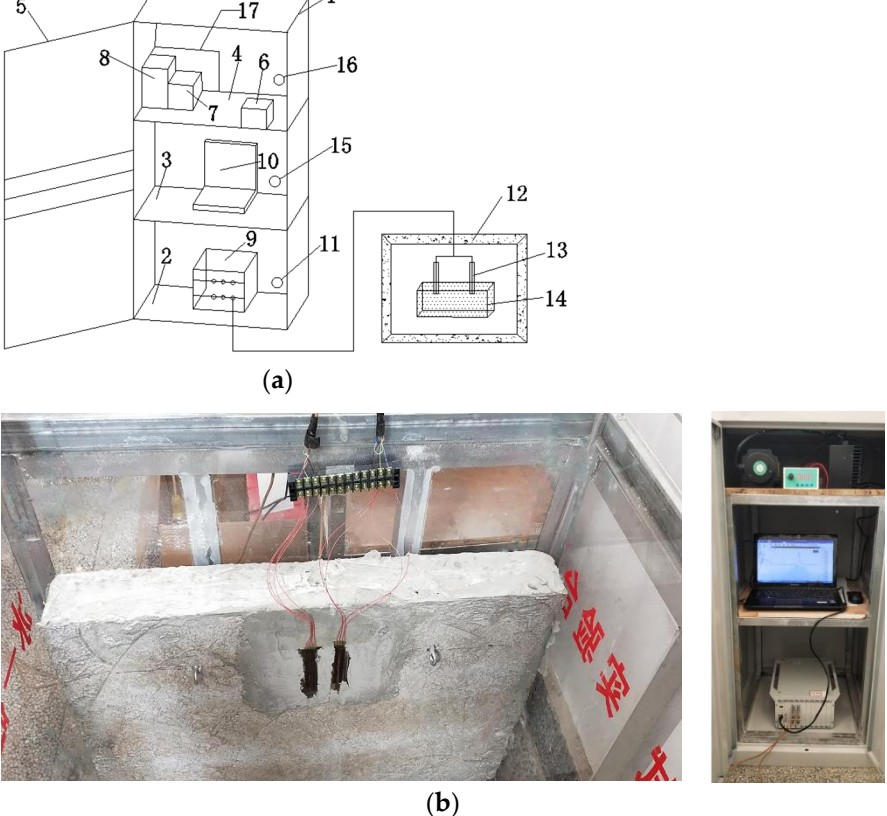

**Figure 3.** (**a**) Test setup diagram. 1. Instrument mounting cabinet. 2. Dynamic strain tester mounting table. 3. Computer mounting table. 4. Power system mounting table. 5. Cabinet doors. 6. Pulse transmitter. 7. Stepper motor driver. 8. Stepper motors. 9. Dynamic strain tester. 10. Computers. 11. Main wiring hole. 12. Concrete lining. 13. Strain gauges. 14. Drift ice. 15. Sub-wiring hole. 16. Wiring holes. 17. Open mouth. (**b**) Test stands and data acquisition and analysis systems.

We prepared a 1 m × 0.12 m × 1 m tunnel lining model with two strain gauges at the waterline of the tunnel lining. A test line was used to connect the strain gauges to the dynamic strain test and analysis system by means of a half-bridge bridge. The computer was started and voltage, modulus of elasticity of concrete, and Poisson's ratio were set in the software. The finished ice model was placed in the system. The power supply system and pulse emitter speed were adjusted, and the dynamic strain curve was recorded in the tunnel lining area as the model ice struck the tunnel lining. After the test had been completed, the data were processed to obtain the peak strain, and the peak stress and peak displacement were then calculated.

After similar scale conversion, the following figure shows the comparison between the simulation and test results of the peak value of equivalent stress and displacement in *X*-direction under different flow ice velocity conditions, as shown in Figure 4.

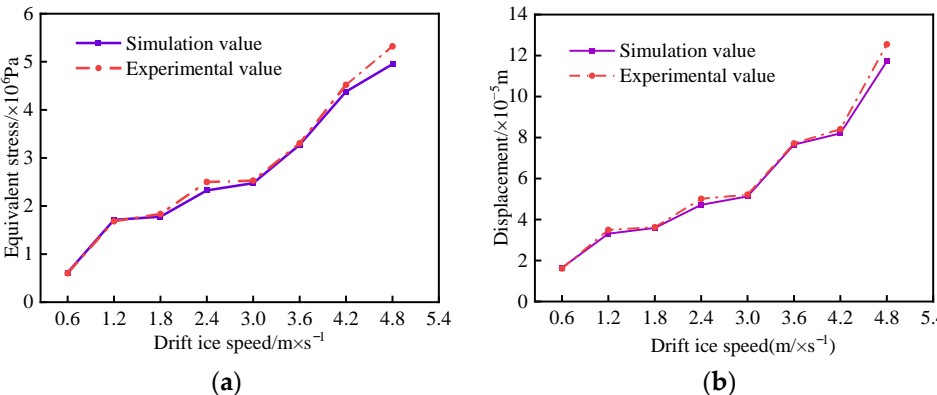

(a)  (b)

**Figure 4.** Comparison of simulation and experimental results of different drift ice speeds. (**a**) Maximum equivalent stress; (**b**) maximum displacement in *X*-direction.

From Figure 4, it can be found that the maximum error between the experimental and simulated values of the peak equivalent stress of tunnel lining under different flow ice velocities is +7.5%, the minimum error is −1.6%, and the average error is +2.58%; the maximum error between the experimental and simulated values of the peak displacement of tunnel lining in *X*-direction under different flow ice velocity is +7.1%, the minimum error is +0.75%, and the average error is +2.88%. The error may be due to the fact that the ice material model chosen for the numerical simulations was an idealized ice model. Second, there were errors in the strain measurement process, the source of which was mainly due to the residual stresses in the concrete lining during the casting of the structure itself and the accuracy of the strain gauges. However, overall, the numerical simulation results and the results obtained from the test values were basically consistent. The data match well, and the error is within the permissible range, indicating that the numerical simulation model meets the accuracy requirements. It can also be seen from Figure 4 that, with an increase in flow ice velocity, both the equivalent stress and peak displacement in *X*-direction show an overall increasing trend. According to the kinetic energy theorem, when the mass of drift ice remains unchanged, as the velocity of drift ice increases, the kinetic energy of drift ice increases and the energy transformed to the tunnel lining structure by the collision energy also increases. This, in turn, causes an increase in equivalent stress and an increase in displacement of the tunnel lining in the *X*-direction. In addition, when the ice speed is greater than 3.0 m/s, the impact of drift ice on the concrete lining of the tunnel is obvious, so engineers should fully consider the impact of drift ice speed on a tunnel when designing the tunnel.

## 4. Numerical Simulation Results and Analysis

### 4.1. Simulation Results under Typical Operating Conditions

In the numerical simulation, the drift ice velocity condition was defined in the interval of 0.6–4.8 m/s, and 3.0 m/s was considered to be the intermediate value. Therefore, 3.0 m/s was accepted for typical working conditions analysis. The selection of the drift ice size considered the local ice conditions during the ice age and the study of flow ice by Xu Guobin [33], and $0.7 \times 0.7 \times 0.3$ m$^3$ was selected for the analysis. At the same time, in order to clarify the influence of the water medium on the collisions, the additional mass model without considering the water medium and the fluid–solid coupling model considering the water medium were established for typical working conditions. The finite element numerical simulation results show that, under the combined working conditions of a drift ice velocity of 3.0 m/s and a drift ice size of $0.7 \times 0.7 \times 0.3$ m$^3$, the maximum equivalent stress cloud map and *X*-direction displacement cloud map of the tunnel lining under the time–history curves is shown.

As can be seen from Figures 5 and 6, (1) Fluid–solid coupling model, the velocity of drift ice is 3.0 m/s and the size of drift ice is $0.7 \times 0.7 \times 0.3$ m$^3$. The maximum equivalent stress in the tunnel lining collision zone and the maximum displacement in the *X*-direction are $2.475 \times 10^6$ Pa and $5.126 \times 10^{-5}$ m, respectively. (2) Additional mass model, the maximum equivalent stress in the tunnel lining collision zone and the maximum displacement in the *X*-direction are $3.960 \times 10^6$ Pa and $9.191 \times 10^{-5}$ m, respectively, under the above combined conditions.

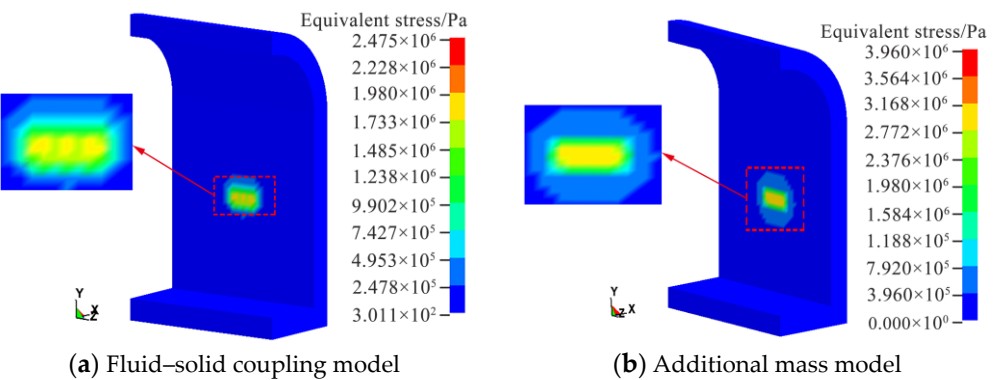

(**a**) Fluid–solid coupling model      (**b**) Additional mass model

**Figure 5.** Equivalent stress clouds under different collision models, where: Local enlargement of the ice−tunnel impact zone is performed.

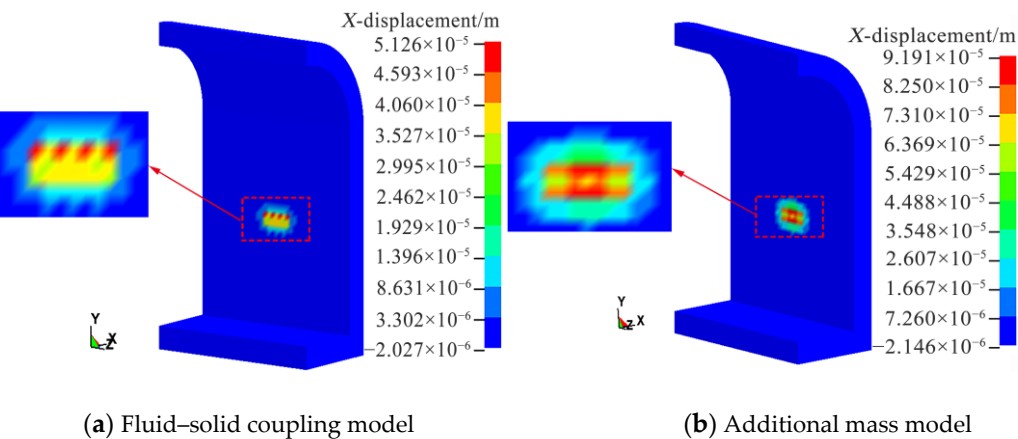

(**a**) Fluid–solid coupling model      (**b**) Additional mass model

**Figure 6.** Displacement clouds in *X*-direction under different collision models, where: Local enlargement of the ice−tunnel impact zone is performed.

As can be seen from Figure 7 (left), (1) Fluid–solid coupling model, the equivalent stress time curve has obvious dynamic nonlinear characteristics and presents a multi-peak

situation. When t = 0~0.002 s, the time course curve of the equivalent stress shows slight fluctuations. The reason for this phenomenon is that the flowing ice hitting the tunnel lining is a gradually approaching process and the water medium between the flowing ice and the tunnel lining will form a high-stress field in advance due to the squeezing effect of the flowing ice movement, which will have an effect on the tunnel lining in advance. In the range of t = 0.00250–0.0305 s, the curve shows an oscillating change with multiple peaks. The reason for this phenomenon is formation of broken ice during the impact process and the influence of waves. After t = 0.00305 s, the equivalent stress is not completely unloaded to zero because the water medium is considered in the calculation. (2) Additional mass model, the dynamic nonlinear characteristics of the equivalent stress–time curve are significantly weaker than those obtained when considering a water medium. The first region of drift ice to come into contact after a drift ice collision will break up first, and the broken ice will fall directly after the ice body breaks up due to the lack of support from the water medium, thus causing a smaller impact. When t = 0~0.000494 s, the equivalent stress–time curve does not change because the collision has not yet occurred. When t = 0.000494~0.0020 s, the drift ice collision begins, and the equivalent stress reaches its peak in a very short time in a nearly straight line, which is shorter than the time taken consider the effect of the water medium. After t = 0.0020 s, the equivalent stress unloads to 0.

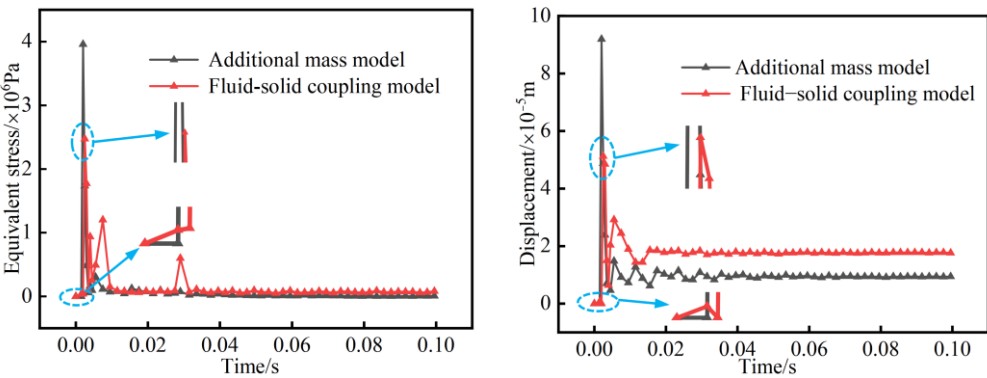

**Figure 7.** Time–history curves of equivalent stresses (**left**) and *X*-displacements (**right**).

As can be seen from Figure 7 (right), (1) Fluid–solid coupling model, at t = 0~0.002 s, the displacement–time course curve of the collision zone shows slight fluctuations. This phenomenon is also due to the influence of the water medium. The reason for this is consistent with the stress–time course curve analysis. When t = 0.00200~0.00250 s, the ice load gradually increases and the *X*-direction displacement of the collision zone shows loading characteristics. The maximum peak occurs at t = 0.00250 s, after which the drift ice bounces back, the ice load gradually decreases, and the *X*-direction displacement in the collision zone shows unloading characteristics. At t = 0.00250–0.0330 s, the displacement in the collision zone shows multiple peaks, indicating that the tunnel lining structure continuously fails after being damaged by the impact. After t = 0.0330 s, the *X*-directional displacement of the tunnel lining stabilizes at around $1.65 \times 10^{-5}$ m. (2) Additional mass model, the trend of the curve is similar to that of the stress–time curve without considering the water medium, and the reasons for this are basically the same. The difference is that, compared to the displacement–time curve under the water medium, the displacement of the tunnel lining in the *X*-direction is stable at around $1 \times 10^{-5}$ m after the collision has been completed.

In order to accurately simulate the changes in stress and displacement at other nodes in the impact zone when the drift ice hits the tunnel, the equivalent stress and displacement curves in the *X*-direction at different locations in the impact zone in the flow–solid coupled model are provided in the figure below.

From Figure 8, it is evident that the equivalent stress time–history curves at different locations in the tunnel impact area have the same variation trend and show nonlinear characteristics; the variation trends of the *X*-direction displacement time–history curves at different locations are basically similar. In addition, it can also be seen from the figure that the *X*-directional displacement of the tunnel lining at different locations is stable at about $1.65 \times 10^{-5}$ m after the collision is completed.

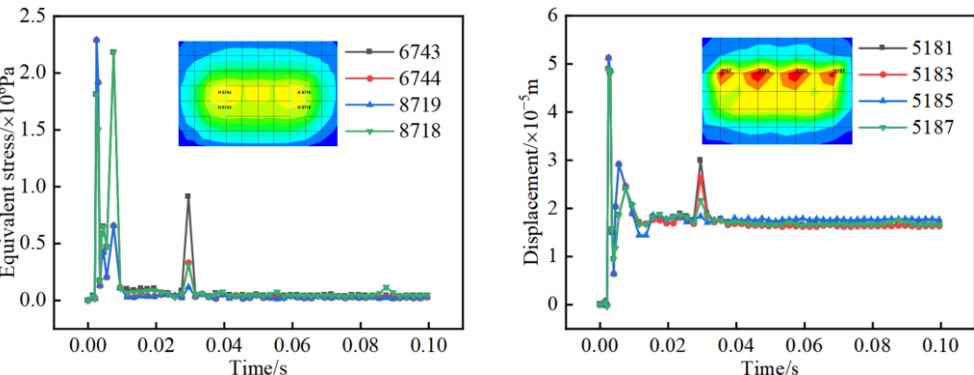

**Figure 8.** Time–history curves of equivalent stresses (**left**) and *X*-displacements (**right**) at different positions.

The following figure further provides the time–history curves of drift ice *x* velocity under different collision models. From Figure 9, the velocity change curves of drift ice in the two different collision models are different. In the fluid–solid coupling model, the velocity of drift ice fluctuates greatly with time due to the action of water body and will gradually return to zero in a short time, while, in the additional mass model, the velocity of drift ice becomes negative instantly after impacting the tunnel liner, i.e., the drift ice is bounced off rapidly, and the velocity remains almost constant in a short time because there is no water medium.

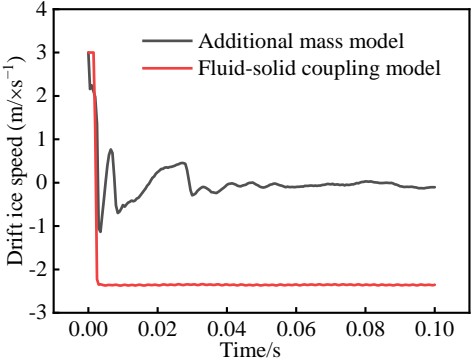

**Figure 9.** Time–history curves of flow ice velocity under different collision models.

## 4.2. Simulation Results for Different Ice Flow Angles

According to the design flow velocity and maximum flow velocity of the tunnel, the velocity of drift ice is 3.0 m/s and the size of drift ice is $0.7 \times 0.7 \times 0.3$ m$^3$. The effect of shear collision between drift ice and tunnel on tunnel lining is investigated. In the finite element simulation, the collision angle$\theta$ between drift ice and tunnel is 0° (positive collision), 15°, 30°, 45°, 60°, and 75°, and the collision angle is shown in Figure 10 ($\theta = 0°$, $\theta = 45°$ for example) without considering secondary collision and other problems.

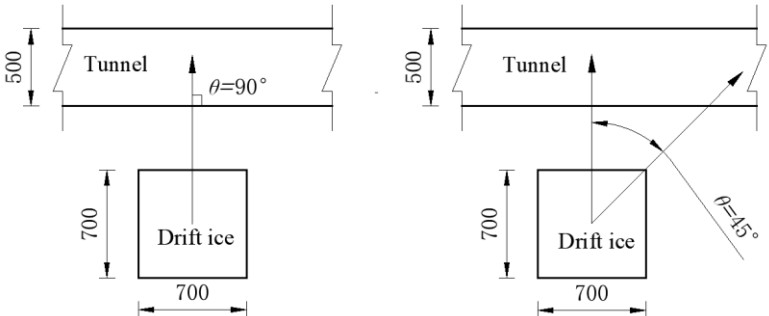

**Figure 10.** Schematic diagram of collision angle between drift ice and tunnel lining (unit: mm).

The following figure provides the time course curves of the equivalent stress of drift ice on tunnel lining and the time course curves of displacement in *X*-direction under different drift ice collision angles. It can also be seen from Figure 11 (right) that, when the collision angle is in the range of 0°–15°, the damage deformation is stable at about $1.6 \times 10^{-5}$ m after the impact is completed; when the collision angle is in the range of 30°–60°, the damage deformation is stable at about $8 \times 10^{-6}$ m; when the collision angle is equal to 75 degrees, the damage displacement fluctuates at 0. It shows that the influence of drift ice on tunnel lining becomes weaker with an increase in collision angle. The influence is most obvious when the collision angle is 0°, mainly because the impact of drift ice on the tunnel lining is greater at the moment of collision and most of the energy of drift ice acts directly on the tunnel lining and only a small part of the energy is absorbed by water medium. With an increase in collision angle, the influence time of water medium on drift ice increases and then consumes more kinetic energy of flow ice.

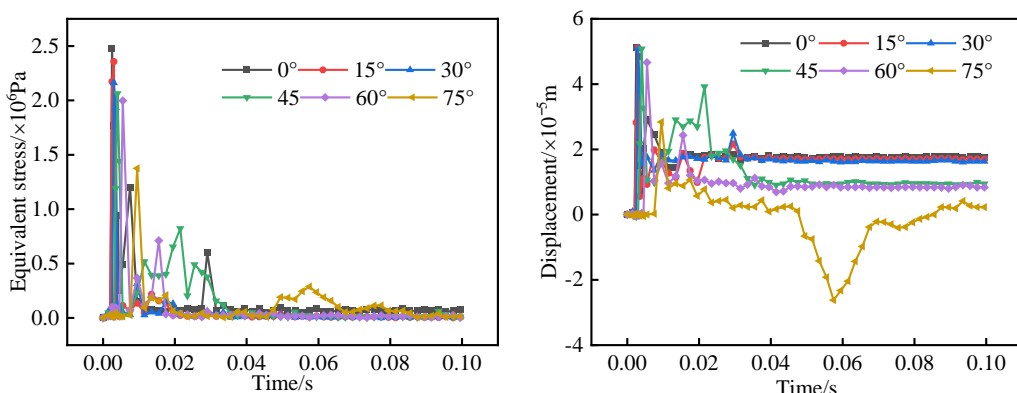

**Figure 11.** Time–history curves of equivalent stresses (**left**) and *X*-displacements (**right**) for different collision angles.

After counting the maximum peaks of the time–history curves in Figure 11, the scatter plots of the peak equivalent stress on the tunnel and the scatter plots of the peak displacement in the *X*-direction under different drift–ice collision angles are plotted and the curves are fitted. The graphs are shown below.

From Figure 12, the peak equivalent stress and the peak displacement in the *X*-direction at different drift ice angles show an exponential function ($y = y_0 + Ae^{(-x/t)}$). The peak equivalent stress and the peak displacement in *X*-direction are decreasing with an increase in drift ice angle. This is similar to the findings of Liu et al. [34]. This indicates that the influence of drift ice angle on the collision of tunnel lining is obvious, so the influence of drift ice angle should be fully considered in the engineering design.

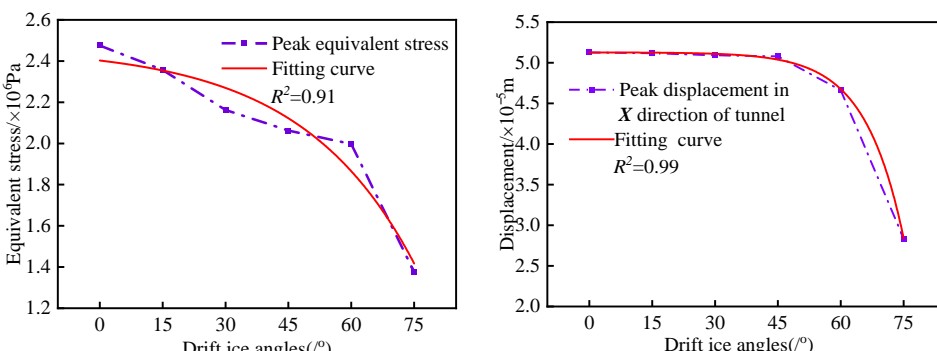

**Figure 12.** Peak graph of Equivalent stress (**left**) and *X*-direction displacement (**right**) for different collision angles.

### 4.3. Simulation Results for Different Drift Ice Sizes

The mechanical properties of drift ice are extremely complex. Jones [35] investigated the effect of ice size on the properties of ice and found that different sizes of ice have certain effects on the mechanical properties and show different "size effects". Therefore, in this paper, considering the ice conditions of Datong River and referring to Xu Guobin's research on drift ice, we selected $0.7 \times 0.7 \times 0.3$ (0.147), $0.7 \times 0.8 \times 0.3$ (0.168), $0.7 \times 0.9 \times 0.3$ (0.189), $0.7 \times 1.0 \times 0.3$ (0.210), $0.7 \times 1.1 \times 0.3$ (0.231), $0.7 \times 1.2 \times 0.3 \text{ m}^3$ (0.252 $\text{m}^3$), and six other conditions to study the "size effect" of drift ice on the ice–tunnel impact. The drift ice velocity was taken to be 3.5 m/s. The following figure provides the equivalent stress and *X*-direction displacement time–history curves for different flow ice size working conditions.

From Figure 13, the time–history of curves of the equivalent stress and time–history curves of displacement in *X*-direction for different sizes of flow ice conditions also show nonlinear characteristics for the same reasons as those analyzed in Section 4.1. It can also be seen from the figure that equivalent stress and peak displacement in the *X*-direction do not occur simultaneously for different size conditions, which is caused by the "water cushion effect" and the "size effect" of the flow ice. Due to the different sizes of drift ice in the process of hitting the tunnel lining, the size of the drift ice increases, the corresponding collision area also increases, and the resistance of the water medium to the drift ice also increases, which, in turn, causes a time difference in the contact between the drift ice and the tunnel lining, resulting in the curve peak not appearing at the same time.

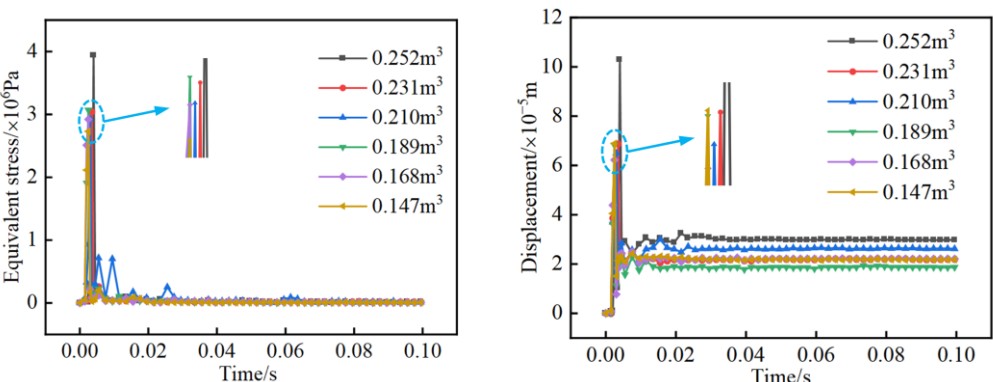

**Figure 13.** Time–history curves of equivalent stresses (**left**) and *X*-displacements (**right**) for different drift ice sizes.

After counting the maximum peaks of the time–history curves in Figure 13, the scatter plots of the peak equivalent stress on the tunnel and the scatter plots of the peak displacement in the *X*-direction for different flow ice sizes were plotted and the curves were fitted. The graphs are shown below.

As can be seen from Figure 14, the drift ice size showed an exponential function ($y = y_0 + Ae^{(-x/t)}$) with respect to both the equivalent stress and the maximum displacement in the *X*-direction. It can also be seen from the figure that the peak equivalent stress and the peak *X*-directional displacement decrease in the range of 0.189–0.210 m$^3$, which is due to the combined effect of the water medium and the "size effect" of the drift ice size. In this paper, the impact of small- and medium-sized drift ice on the tunnel lining is studied, and the increase in drift ice size increases the mass of drift ice relatively little, so the differences in peak equivalent stress and *X*-direction displacement of different sizes of drift ice are correspondingly small. In addition, with an increase in drift ice size, the resistance of water medium to drift ice increases accordingly, the slowing down effect on drift ice velocity increases, and the impact area between drift ice and tunnel lining also increases accordingly, so the equivalent stress and peak *X*-direction displacement of tunnel lining will show a slight downward trend instead of a linear growth trend. When the drift ice size is larger than 0.7 × 1.0 × 0.3 (0.210 m$^3$), the peak equivalent stress and peak *X*-direction displacement both increase with an increase in drift ice size. The above phenomenon shows that, when the drift ice size is smaller than 0.7 × 1.0 × 0.3 (0.210 m$^3$), the "size effect" of water medium and drift ice size has obvious influence on it, while, when the drift ice size is larger than 0.7 × 1.0 × 0.3 (0.210 m$^3$), the "size effect" of the water medium and size of flow ice diminish.

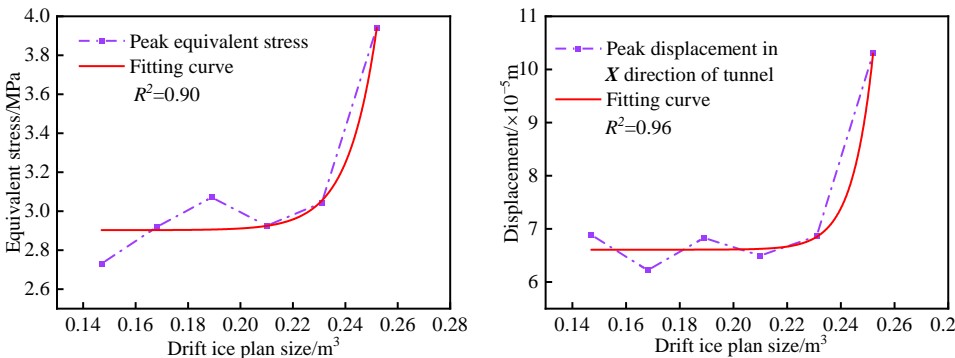

**Figure 14.** Peak graph of Equivalent stress (**left**) and *X*-direction displacement (**right**) for different drift ice sizes.

### 4.4. Simulation Results for Different Tunnel Cross-Sections

According to research on the cross-sectional form of water conveyance tunnel conducted by Wang Cuncun [36], at present, the main cross-sectional forms of a water conveyance tunnel are the city gate, horseshoe, and round tunnels. Therefore, models of the water conveyance tunnel under the above three cross-sectional forms were established in this study. This study further analyzed the influence of drift ice on the lining of the water conveyance tunnel under different cross-sectional forms. Based on the analysis of the simulation results under different drift ice sizes shown in Section 4.3, this section mainly discusses the simulation results for different tunnel sections. In order to compare and analyze the influence of size change of impact body (drift ice) and shape change of impact body (tunnel) on the tunnel lining, the speed was kept consistent and taken as 3.5 m/s, and the size of drift ice was taken as 0.7 × 0.7 × 0.3 m$^3$ during the process. Using the combined working conditions of the above two factors, the *X*-direction displacement and maximum equivalent stress of the tunnel lining under different cross-sectional forms are shown in the following figure.

Figure 15a,b shows that the maximum equivalent stress on the lining of Shing Mun Tunnel is 2.73 × 10$^6$ Pa, and the maximum displacement in the *X*-direction is 6.89 × 10$^{-5}$ m. Similarly, according to Figures 16 and 17, the maximum equivalent stress values of the tunnel lining under the horseshoe shape and circular section are 2.34 × 10$^6$ Pa and 1.53 × 10$^6$ Pa, respectively, and the maximum displacement values in the *X*-direction are 4.24 × 10$^{-5}$ m and 3.12 × 10$^{-5}$ m, respectively. Further, when drift ice hits the tunnel

lining, the city gate type has the greatest influence on the tunnel lining, with the horseshoe shape having the second greatest influence, and the circular section has the least influence. The equivalent stress time–history curve and the *X*-direction displacement time–history curve of drift ice on the tunnel lining under different cross-sectional forms are shown in the figure.

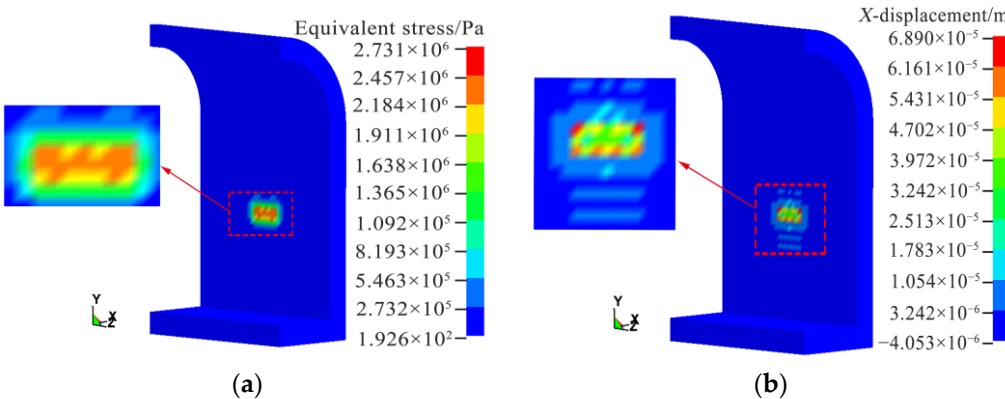

**Figure 15.** Cross section of city gate tunnel. (**a**) Equivalent stress peak value. (**b**) Peak displacement in *X*-direction.

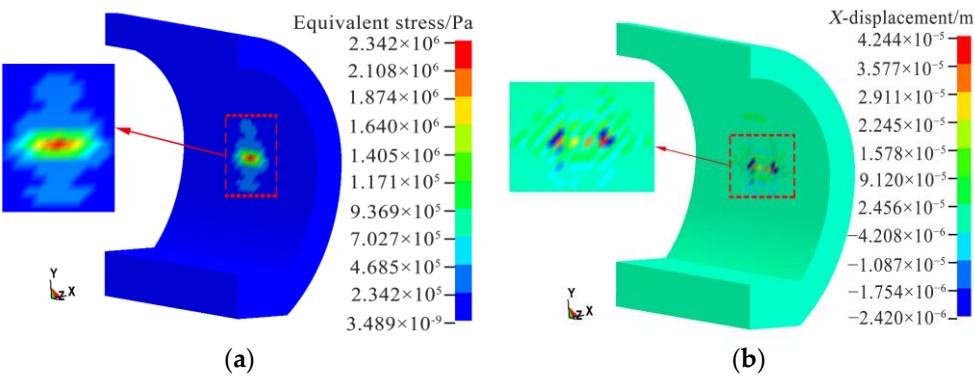

**Figure 16.** Horseshoe tunnel section. (**a**) Equivalent stress peak value. (**b**) Peak displacement in *X*-direction.

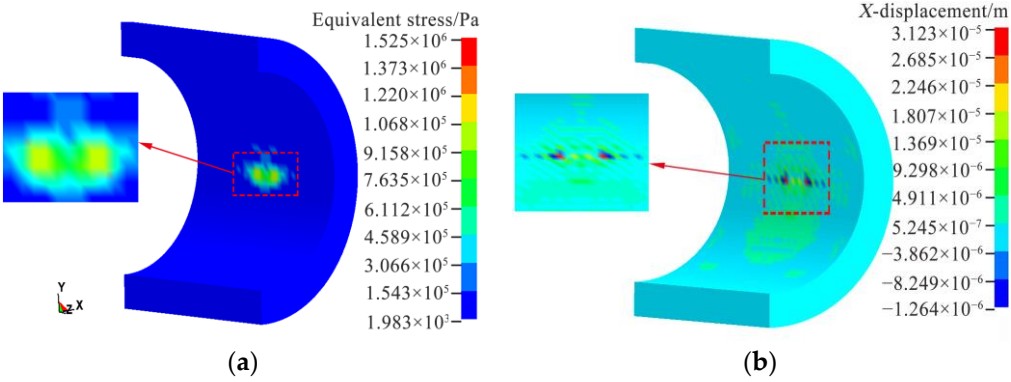

**Figure 17.** Circular tunnel section. (**a**) Equivalent stress peak value. (**b**) Peak displacement in *X*-direction.

As can be seen from Figure 18, the trajectories of the equivalent stress and *X*-direction displacement curves for the horseshoe and circular cross-sectional forms are similar, both showing obvious nonlinear characteristics. The nonlinear characteristics of the equivalent stress and *X*-directional displacement time curves are weaker in the Shing Mun form. The

maximum equivalent stress and maximum displacement in the *X*-direction decrease as the curvature of the tunnel structure increases, indicating that, the greater the curvature of the tunnel structure, the better its crashworthiness. It can also be seen from Figure 18 (right) that the drift ice causes some damage to the tunnel lining in all three different section forms, with the damage deformation in the Shing Mun section form stabilizing at around $2.18 \times 10^{-5}$ m after impact and the horseshoe and circle showing fluctuations around 0. The reason for this phenomenon is that, because the curvature of the city gate tunnel structure is 0, with deepening of the drift ice impact, the water medium is completely squeezed out, whereas the horseshoe and circular tunnel structures have a certain curvature, so the water medium does not completely escape when the drift ice impacts, thus causing the curve form to fluctuate. The water medium that does not escape plays a certain role in blocking the drift ice, resulting in a significantly smaller impact on the tunnel lining than in the Shing Mun Tunnel section. To sum up, it is particularly important to select a reasonable tunnel cross-section according to the characteristics of regional drift ice hazards during engineering design.

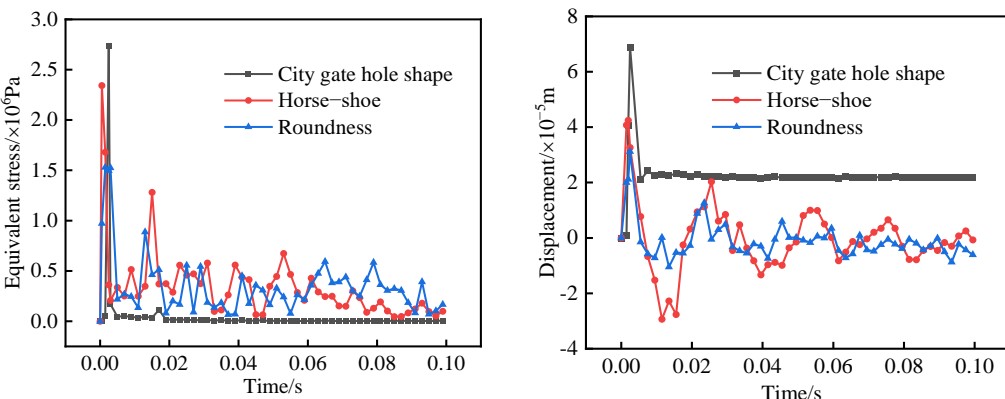

**Figure 18.** Time–history curves of equivalent stresses (**left**) and *X*-displacements (**right**) for different cross-sections.

The average peak of the equivalent stress when the drift ice size changes is 1.53 times the average peak of the different tunnel section forms and 1.41 times the average peak of the *X*-directional displacement, indicating that the effect of the change in drift ice size on the tunnel lining is greater than the effect of the tunnel section forms.

In order to further clarify the influence of water medium on ice–tunnel collisions under different sections, the drift ice velocity during the collision was analyzed, as can be seen from Figure 19, When t = 0~0.00499 s, the drift ice velocity under the three sections showed a linear decrease. The reason for this phenomenon is that the drift ice hitting the tunnel lining is a gradually approaching process, and the water medium between the drift ice and tunnel lining will form a high-stress field in advance due to occurrence of the extrusion effect during drift ice movement; this, in turn, has an impact on the drift ice movement, and the distance between the ice and tunnel was set at 0.005 m, so the drift ice velocity decreased more rapidly. For t = 0.00499~0.05849 s, part of the kinetic energy of the drift ice is transformed into the tunnel lining during the ice–tunnel collision, part of the deformation energy of the drift ice is transformed into water energy and hourglass energy, and the remaining portion of energy is retained as kinetic energy of the drift ice after the collision. As can be seen from the law of conservation of energy, the velocity of the drift ice will not rebound to the initial velocity after the end of the collision. It can also be seen from the figure that the curve shape changes significantly during this time period due to fluctuations caused by perturbations in the water medium. After t = 0.05849 s, the velocity of the drift ice under all three section forms fluctuates around 0 m/s, but, overall, the velocity of the drift ice under the circular section form is greater than that under the horseshoe and Shing Mun cave forms. The reason for this phenomenon is that, due to the

circular tunnel structure having the greatest curvature, when the drift ice hits the tunnel, the water medium between the ice and the tunnel is stored to a greater extent, so the drift ice bounces back more with greater repulsive force.

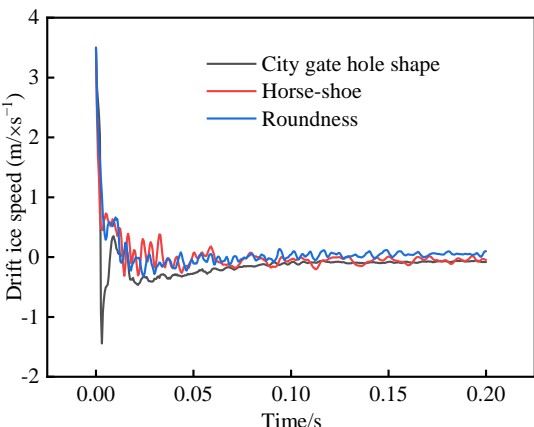

**Figure 19.** Velocity time–history curves of flowing ice under different cross-sections.

## 5. Conclusions

In this paper, based on nonlinear finite element numerical simulation software, collision simulation of drift ice on water transfer tunnel lining with different drift ice collision angles, drift ice sizes, and tunnel section forms was carried out, and corresponding model tests were carried out for validation. The following conclusions were obtained:

(1) The experimental values of maximum equivalent stress and *X*-directional displacement show the same trend as the simulated values by comparing with the experimental results of the model of impact of drift ice on the water transfer tunnel, both of which show an increasing trend with an increase in drift ice velocity. This also indicates that the ice material model parameters, ALE algorithm, and grid size used in this paper can simulate the impact of drift ice on the water transfer tunnel more accurately. In addition, it is found that, when the ice velocity is greater than 3.0 m/s, the impact of drift ice on the water transfer tunnel is obvious, so engineers should fully consider the impact of drift ice velocity on a tunnel when designing the tunnel.

(2) Comprehensive analysis of the different drift ice conditions simulated in this study shows that, as drift ice collision angle and drift ice size increase, the fitted curves of equivalent stress and peak displacement in *X*-direction all show an exponential function. The difference is that equivalent stress and peak displacement in *X*-direction keep decreasing as drift ice collision angle increases, while the most obvious effect of positive drift ice collision on the tunnel lining is found. However, as size of drift ice increases, peak equivalent stress and *X*-directional displacement show different trends, and peak equivalent stress and *X*-directional displacement decrease in the range of 0.189–0.210 m$^3$, indicating that the "size effect" of drift ice and the influence of water medium on the ice–tunnel collision process are obvious. When the drift ice size is larger than 0.210 m$^3$, the peaks all increase with an increase in drift ice size, indicating that "size effect" and water medium have a significantly lower influence at this time. In the case of changing only the tunnel section form, both the peak equivalent stress and peak displacement in the *X*-direction in the tunnel lining impact zone decrease as the curvature of the tunnel structure increases, indicating that, the greater the curvature of the tunnel structure, the better its crashworthiness. In addition, a comparative analysis of the effect on tunnel lining under different drift ice sizes and tunnel section forms was carried out and it was found that the average peak equivalent stress when the drift ice size changes is 1.53 times higher than the average peak value of different tunnel section forms and 1.41 times higher than the average peak *X*-directional displacement. Therefore, regional drift ice disaster characteristics

in the engineering design should fully consider drift ice collision angle, drift ice size, and tunnel cross-section form on the impact of a water transmission tunnel.

(3) Through comparative analysis of the water medium under typical working conditions by the additional mass method and fluid–solid coupling method, it was found that a high-pressure field will be formed by extrusion of the water medium during drift ice movement. This will have an impact on the tunnel lining during the impact process, and, through analysis of drift ice velocity under different tunnel sections, it was found that water medium has an obvious influence on drift ice movement. This should be fully considered in numerical simulations.

**Author Contributions:** L.G.: Methodology, Software, Investigation, Data curation, Funding acquisition, Writing—original draft, Visualization. Z.D.: Methodology, Supervision, Validation, Resources, Writing—review and editing. C.J.: Conceptualization, Methodology, Supervision, Validation, Resources, Writing—review and editing. Z.J.: Methodology, Supervision, Resources, Writing—review and editing. T.Y.: Methodology, Software, Supervision, Writing—review and editing. All authors have read and agreed to the published version of the manuscript.

**Funding:** This study was the financially supported by the National Science Foundation (51969011, 72261024) of China.

**Data Availability Statement:** All data included in this study are available upon request by contacting the corresponding author.

**Conflicts of Interest:** The authors declare no conflict of interest.

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
