# Peer review of "Flow–Solid Coupling Analysis of Ice–Concrete Collision Nonlinear Problems in the Yellow River Basin"

_water, doi:10.3390/w15040643_

Round 1
Reviewer 1 Report
The Yellow River ice is the most prominent and significant natural disaster in winter and spring in China. The paper uses ANSYS/LS-DYNA software to perform a non-linear finite element simulation of the impact process between drift ice and tunnel in a water-air coupled medium, and the validity of the numerical simulation results is verified through model tests. The dynamic response of drift ice impacting the water transfer tunnel is investigated using the angle of collision between drift ice and tunnel, drift ice size, and tunnel cross-sectional form as variables.
This is an interesting research work and its findings have some engineering significance and practical value for the prevention and control of ice disasters in the Yellow River Basin. I suggest to be accepted after minor revision, the following issues need to be further improved:
Minor comments:
1. In the description of the water and air medium materials in section 3.2, the authors only give the sources of the parameters of the intrinsic model and suggest that the authors add the sources of the parameters of the equations of state for the water and air mediums.
2. From the description of the ice material in section 3.2 it can be seen that in the numerical simulation the ice material was selected as *MAT_STRAIN_RATE_DEPENDENT_PLASTICITY. is there some basis for the selection of the flow ice material parameter in the text? It is suggested that the authors add an appropriate description.
3. When building the finite element model, the contact algorithm used by the authors is the symmetric penalty function method and the contact type used is the automatic surface-to-surface contact (ASSC). It is suggested that the authors add the corresponding literature support.
4. In Figure 2 in section 3.3, why is only a smaller part of the air medium modeled and how is the air medium in the water part of the tunnel considered? It is recommended that the authors add an appropriate description.
5. As can be seen from Figure 4 in Section 3.4, there is some error in the results of the model tests and numerical simulations for different drift ice velocity operating conditions. However, the authors do not elaborate on the source of the errors and it is recommended that the authors add an appropriate explanation. In addition, the authors are requested to further check the labeling of Fig. 4(a).
6. Section 4.1 describes the role of the water-air coupling medium, but Figs. 7 and 8 do not show it visually, and it is suggested that the authors add a local enlargement at t=0~0.002s. In addition, an extra space has been added in line 357 for "5.13 × 10-5", please delete it.
7. It is suggested that the authors add whether the drift ice impacting the tunnel lining caused cracking of the tunnel lining.
8. As can be seen from section 4, the paper is mainly written within the framework of different drift ice collision angles, different drift ice sizes, and different tunnel section forms. However, in the conclusion and abstract, the authors write the book according to the logic of different drift ice dimensions, different drift ice collision angles, and different tunnel section forms. It is suggested that the authors adjust the order of variables in the abstract and conclusion to be consistent with section 4 in order to enhance the logic of the paper.
Author Response
请参阅附件

Reviewer 2 Report
Please see the attached file.

Reviewer 3 Report
Generally, based on a finite element model and using ANSYS/LS-DYNA the manuscript discussed the collision process between ice and the concrete wall of the tunnel. To verify the numerical modeling, an experimental setup is used. Then the numerical model is run for different conditions and scenarios such as typical operation, different ice flow angles, different drift ice sizes, and different tunnel cross-sections. The research topic investigated is interesting, however, there are cases in the manuscript that should be addressed. Therefore, the manuscript should be revised with major corrections.
1- Introduction. Since the water transmission tunnel generally works under pressurized and free surface conditions along the tunnel path, this structure is faced different types of interaction issues such as fluid-structure interaction, soil/rock-structure interaction, and hydromechanical subjects. Adding some previous numerical studies related to those engineering challenges will enrich this section further. For example:
- Investigating the effects of transient flow in concrete-lined pressure tunnels and developing a new analytical formula for pressure wave velocity, published in Tunnelling and Underground Space Technology 2019.
- Mechanical-Hydraulic Interaction in the Cracking Process of Pressure Tunnel Linings, published in International Journal on Hydropower and Dams 2013.
- Stress intensity factors for axial semi-elliptical surface cracks and embedded elliptical cracks at longitudinal butt welded joints of steel-lined pressure tunnels and shafts considering weld shape, published in Engineering Fracture Mechanics, 2017
2- Equations 7, 8, and 9– Please clarify what is w?
3- Table 2- The title of the table should be concrete properties.
4- Table 3- What are constants C, S1, S2, C4, C5 for water and air?
5- 3.3 Model building. Please clarify the boundary conditions of tunnel lining.
6- Figure 4-b. Since the plastic behavior is assigned to the concrete to study the damage caused by the collision of ice, please clarify why the authors used displacement values for investigations, however, the failure criteria for concrete as brittle material should be considered. Also, the range of displacements is too low (order of magnitude 10-5), even ignorable in engineering design.
7- Figures 5-a and 6-c. The legends reflect the maximum stresses of 2.47 MPa and 3.96 MPa respectively with a red contour, however, there is no such zone in the lining.
8- Figures 5-b and 6-d. The concrete lining has a negative displacement (-2×10-6), what is this negative displacement? Also, it seems that the location of maximum displacement is not corresponding to the maximum stresses.
9- Figure 6. Captions should be corrected to a and b.
10- Figures 12 and 13. According to the results (purple line), it is difficult to say there is a correlation between the size of the ice and stress and displacement. How did the authors conclude that the size of the ice has a parabolic correlation (polynomial) with Stress and displacement?
Round 2
Reviewer 2 Report
Dear authors,
you have improved significantly the quality of the manuscript by correcting issues, adding clarifications and generating additional material (e.g. tables and figures). This new material is very useful for the reader. Therefore, you have addressed properly my comments and the manuscript is suggested for publication.
My only advice would be to include in the actual manuscript (e.g. in the main text OR in an Appendix) the table about the mesh sensitivity (see below), which you showed in your reply for comment 5. Including the Table below, will convince all the readers of the manuscripts about the accuracy of your results.
Response: Three sets of grids A1, A2 and A3 were selected with corresponding dimensions of 0.05×0.05×0.05, 0.08×0.08×0.08 and 0.10×0.10×0.10 m3, and the peak equivalent stress and peak displacement in the X-direction derived from the numerical calculations at different grid scales are given in Table 1.
Table 1 Comparison of peak equivalent stress and peak displacement in X-direction at different grid scales
|
mesh serial number |
A1 |
A2 |
A3 |
Experimental value |
|
mesh size |
0.05×0.05×0.05m3 |
0.08×0.08×0.08m3 |
0.10×0.10×0.10m3 |
|
|
Peak equivalent stress(×106Pa) |
2.475(2.1%) |
2.916(-15%) |
3.211(-27%) |
2.528 |
|
X-direction displacement peak(×10-5m) |
5.216(0%) |
5.913(-13%) |
6.295(-21%) |
5.216 |
From Table 1, we can see that the errors of the experimental values of the peak equivalent stress obtained from the simulations of A1, A2 and A3 meshes are 2.1%, -15% and -27%, respectively; the errors of the peak displacement in the X-direction obtained from the simulations of A1, A2 and A3 meshes and the experimental values are 0%, -13% and -21%, respectively. Through the above above mesh size sensitivity analysis, the A1 (0.05×0.05×0.05m3 ) mesh is used in this paper for the following numerical calculations.
Reviewer 3 Report
In my opinion, the present version of this manuscript can be accepted since the Authors addressed my requests properly.